# FoldMold: Automating Papercraft for Fast DIY Casting of Scalable Curved Shapes

Hanieh Shakeri *       Hannah Elbaggari †       Paul Bucci ‡       Robert Xiao §       Karon E. MacLean ¶

Department of Computer Science
University of British Columbia

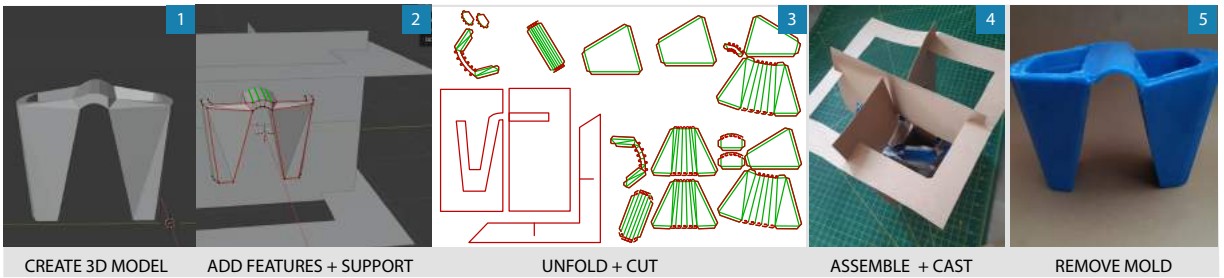

| | | | | |
|---|---|---|---|---|
| CREATE 3D MODEL | ADD FEATURES + SUPPORT | UNFOLD + CUT | ASSEMBLE + CAST | REMOVE MOLD |

Figure 1: FoldMold enables rapid casting of curved 3D shapes using paper and wax. In this image we show the steps for casting a silicone kitchen gripper, far faster and with less waste than it can be 3D printed. (1) Create a 3D model of the object positive in Blender. (2) Add joinery and support features using the FoldMold Pattern Builder. (3) Automatically unfold the model into 2D patterns, and cut them from paper. (4) Assemble the mold and pour the intended material. (5) Remove the cast object from the mold.

## ABSTRACT

Rapid iteration is crucial to effective prototyping; yet making certain objects – large, smoothly curved and/or of specific material – requires specialized equipment or considerable time. To improve access to casting such objects, we developed FoldMold: a low-cost, simply-resourced and eco-friendly technique for creating scalable, curved mold shapes (any developable surface) with wax-stiffened paper. Starting with a 3D digital shape, we define seams, add bending, joinery and mold-strengthening features, and "unfold" the shape into a 2D pattern, which is then cut, assembled, wax-dipped and cast with materials like silicone, plaster, or ice. To access the concept's full power, we facilitated digital pattern creation with a custom Blender add-on. We assessed FoldMold's viability, first with several molding challenges in which it produced smooth, curved shapes far faster than 3D printing would; then with a small user study that confirmed automation usability. Finally, we describe a range of opportunities for further development.

**Index Terms:** Human-centered computing; Human-centered computing—Systems and tools for interaction design

## 1 INTRODUCTION

The practice of many designers, makers, and artists relies on rapid prototyping of physical objects, a process characterized by quick iterative exploration, modeling and construction. One method of bringing designs to life is through *casting*, wherein the maker pours material into a mold "negative" and lets it set [30]. Casting advantages include diversity of material, ability to place insets, and the possibility of mixing computer-modeled and extemporaneous

---
*e-mail:haniehs@cs.ubc.ca

†e-mail:hre@cs.ubc.ca

‡e-mail:pbucci@cs.ubc.ca

§e-mail:brx@cs.ubc.ca

¶e-mail:maclean@cs.ubc.ca

manual mold construction [38].

A primary downside to casting for rapid prototyping is the time required to build a mold. 3D printing a mold (a common approach today) can take hours even for small objects. For multi-part molds, large models and with failed prints, the latency grows to days. While other mold-making techniques exist (*e.g.*, *StackMold* [49], *Meta-molds* [4]), they are either limited to geometrically simple shapes such as extrusions, or require effort in other ways. Molds of smooth and complexly curved, digitally defined models are hard to achieve in a Do-It-Yourself (DIY) setting except through 3D printing. Iteration is thus expensive, particularly for large sizes and curvy shapes.

Efficient use of materials is a particular challenge for prototype casting. When iterating, we only need one-time-use molds; so both the mold and final object materials should not only be readily available and low-cost, but also minimize non-biodegradable waste – a problem with most 3D printing media [26].

This work was inspired by insights connecting papercraft and computer-aided fabrication. The first was that papercraft and wax can together produce low-cost, eco-friendly, curved molds. Papercraft techniques like origami (paper folding) and kirigami (paper cutting) [13, 19] can produce geometrically complex *positive* shapes. Because paper is thin, the *negative* space within can be filled with castable material for new positive objects. Paper can be flexibly bent into smooth curves and shapes. Mold construction time is size invariant. *Wax* can fix, reinforce, and seal a paper mold's curves. It is biodegradable, easy to work with, melts at low heat, creates a smooth finish, is adhesive when warm, can be iteratively built and touched-up. Both paper and wax are inexpensive and easy to source.

Secondly, while origami and kirigami shapes are folded from paper sheets, paper pattern pieces can also be *joined* with woodcraft techniques. Paper has wood-like properties that enable many cutting and assembly methods: it is fibrous, tough, and diverse in stiffness and density. Like wood, paper fibers allow controlled *bending* through patterned cuts. Unlike wood, its weakness can be exploited for bending, and mold parts are easily broken away after casting.

Thirdly, the complex design of paper and wax molds is highly automatable. Given user-defined vertices, edges, and faces, we can algorithmically compute mold structure, bends, joints and seams, and mold supports – steps which would require expertise and time,

especially for complex shapes. With a computational tool, we can make the pattern-creation process very fast, more precise, and reliable, while still allowing a maker's intervention when desired.

**FoldMold** is a system for rapidly building single-use molds for castable materials out of wax-stiffened paper that blends paper-bending and wood joinery methods (Figure 1). To support complex shapes, we created a computational tool – the *FoldMold Pattern Builder* (or "*Builder*") – to automate translation of a 3D model to a 2D pattern with joinery and mold support components. Patterns can be cut with digital support (*e.g.*, lasercut or, at home, with X-acto knife or vinyl cutter [1]).

In this paper, we show that FoldMolds are faster to construct than other moldmaking methods, use readily-available equipment and biodegradable materials, are low-cost, support complex shapes difficult to attain with other methods, and can be used with a variety of casting materials. FoldMold is ideal for custom fabrication and rapid iteration of shapes with these qualities, including soft robotics, wearables, and large objects. FoldMold is valuable for makers without access to expensive, high-speed equipment and industrial materials, or committed to avoiding waste and toxicity.

## 1.1 Objectives and Contributions

We prioritized three attributes in a mold-making process:

*Accessibility:* Mold materials should be cheap, accessible, and disposable/biodegradable. The pattern should be easy to cut (*e.g.*, laser or vinyl cutter) and assemble in a typical DIY workshop.

*Speed and Outcome*: Moldmaking should be fast, support high curve fidelity and fine surface finish, or be a useful compromise of these relative to current rapid 2D-to-3D prototyping practices.

*Usability and Customisability:* Mold creation and physical assembly should be straightforward for a DIY maker, hiding tedious details, yet enabling them to customize and modify patterns.

To this end, we contribute:

1. The novel approach of saturating digitally-designed papercraft molds with wax to quickly create low-cost, material-efficient, laser-cuttable molds for castable objects;

2. A computational tool that makes it feasible and fast to design complex FoldMolds;

3. A demonstration of diverse process capabilities through three casting examples.

## 2 RELATED WORK

We ground our approach in literature on prototyping complex object positives, in rapid shape prototyping, casting, papercraft and woodcraft techniques, and computational mold creation.

### 2.1 Rapid Shape Prototyping

#### 2.1.1 Additive Prototyping

Based on sequentially adding material to create a shape, additive methods are dominated by 3D printing [20] due to platform penetration, slowly growing material choices, precision and resolution, and total-job speed and hands-off process relative to previous methods such as photo sculpture [45] and directed light fabrication [29]. 3D printers heat and extrude polymer filaments. Some technologies can achieve high resolution and precision at the expense of speed, cost and material options [36].

Rates of contemporary 3D printing are still slow enough (hours to days) to impede quick iteration. As an example of efforts to increase speed, *WirePrint* modifies the digital 3D model to reflect a mesh version of the object positive [33], but at the cost of creating discontinuous object surfaces. While capturing an object's general shape and size, it sacrifices fidelity. Other limitations of direct 3D printing of object positives are limited material options and geometry constraints. Its layering process complicates overhangs: they require printed scaffolding, or multi-part prints for later reassembly [23]. Papercraft has no problem with overhangs; where mold support is needed FoldMold utilizes paper or cardboard scaffolding stiffened with wax.

#### 2.1.2 Subtractive and 2D-to-3D Prototyping

Computer Numerical Control (CNC) machining technologies include drills and laser-cutters, and lathes and milling machines which can create 2D and 3D artifacts respectively, all via cutting rather than building-up. Although limited to 2D media, laser-cutting offers speed and precision at low per-job cost, albeit with a high equipment investment [39]. Because it can be cheaper and faster to fabricate 2D than 3D media, some have sought speed by cutting 2D patterns to be folded or assembled into 3D objects [7, 8, 34].

*FlatFitFab* [28] and *Field-Aligned Mesh Joinery* [14] allow the user to create 2D laser cut pieces that, when aligned and assembled, form non-continuous 3D approximations of the object positive - essentially creating the object "skeleton". Other methods (*e.g.*, *Joinery* [52], *SpringFit* [42]) utilize laser cutting followed by assembly of 2D cutouts. *Joinery* supports the creation of continuous, non-curved surfaces joined by a variety of mechanisms. *SpringFit* introduces the use of unidirectional laser-cut curves joined using stress-spring mechanisms. However, these techniques are for creating object positives and not suitable for casting; material qualities are inherently limited and the joints are not designed to fully seal.

Here, we draw inspiration from these methods which approach physical 3D object construction based on 2D fabrication techniques, and draw on the basic ideas to build continuous sealed object negatives (molds) for casting objects from a variety of materials.

### 2.2 Casting in Rapid Prototyping

An Iron Age technique [30], casting enables object creation through replication (creating a mold from the target object's positive) or from designs that do not physically exist yet (our focus). A particular utility of casting in prototyping is access to a wider range of materials than is afforded by methods like 3D printing, carving or machining – *e.g.*, silicone or plaster. For example, Babaei et al. [6] employ clear molds to cast photopolymer objects.

*StackMold* is a system for casting multi-material parts that forms molds from stacked laser-cut wood [49]. It incorporates lost-wax cast parts to create cavities for internal structures. While this improves casting speed (especially with thicker layers), the layers create a discretized, "stepped" surface finish which is unsuitable for smoothly curved shapes – prototyping speed is in conflict with surface resolution. *Metamolds* [4] uses a 3D printed mold to produce a second silicone mold, which is then used to cast objects. The Metamolds software minimizes the number of 3D printed parts to optimize printing time. Silicone molds are good for repeated casts of the same object, but this multi-stage process slows rapid iteration requiring only single-use molds. Further, Metamolds are size-constrained by the 3D printer workspace.

Thus, despite significant progress in rapid molding, fast iteration of large and/or complex shapes is still far from well supported.

### 2.3 Papercraft and Wood Modeling

Several paper and wood crafting techniques inspired FoldMold.

#### 2.3.1 Papercraft

*Origami* involves repeatedly folding a single paper sheet into a 3D shape [12]. Mathematicians have characterized origami geometries [32] as Euclidean constructions [19]. They can achieve astonishing complexity, but at a high cost in labor, dexterity and ingenuity. *Kirigami* allows paper cutting as well as folding to simplify assembly and access a broader geometric range. Despite the effort, both demonstrate how folding can transform 2D sheets into complex 3D shapes, and that papercraft design can be modeled.

### 2.3.2 Creating 2D Papercraft Patterns for 3D Objects

Many have sought ways to create foldable patterns and control deformation by discretizing 3D objects. Castle et al. [11] developed a set of transferable rules for folding, cutting and joining rigid lattice materials. For 3D kirigami structures, specific cuts to flat material can be buckled out of plane by a controlled tension on connected ligaments [40]. Research work on these papercraft techniques inform cut and fold prototyping systems; *e.g.*, *LaserOrigami* uses a laser cutter to make cuts on a 2D sheet then melts them into specified bends for a precise 3D object [34]. FoldMold goes beyond this by enabling the use of a wide variety of materials through casting, and supporting the creation of large, curvy objects.

### 2.3.3 Controlled Bending

Wood and other rigid but fibrous materials can be controllably bent with partial cuts, by managing cut width, shape and patterning [48]. Many techniques and designs achieve specific curves: *e.g.*, *kerfing*, patterns of short through-cuts, can render a different and more continuous curvature than *scoring* (cutting partway through) [10,22,31,51]. These methods support complex double curved surfaces [10, 31], stretching [21], and conformation to preexisting curves for measurements [50]. With sturdy 2D materials, they create continuous curves strong enough to structurally reinforce substantial objects [8].

### 2.3.4 Joinery

In fine woodwork, wood pieces are cut with geometries that are pressure-fit into one another, to mechanically strengthen the material bond which can be further reinforced with glue, screws or dowels. There are many joint types varying in ideal material and needed strength. Taking these ideas into prototyping, *Joinery* developed a parametric joinery design tool specifically for laser cutting to create 3D shapes [52]. Joinery has been used in rapid prototyping literature: Cignoni et al. [14] creates a meshed, interlocked structure approximation of a positive shape to replicate a 3D solid object. Conversely, *SpringFit* shows how mechanical joints can lock components of an object firmly in place and minimize assembly pieces [42].

Our work leverages these papercraft, joinery and modeling techniques to achieve structurally sound, complex and curved shapes from 2D materials by using precise bending and joints.

## 2.4 Computational Mold Creation

Computational support can make complex geometric tasks more accessible to designers [9, 15, 25, 37, 41, 43, 44]. *LASEC* allows for simplified production of stretchable circuits through a design software and laser cutter [21]. Some of these tools include software that automates part of the process [27]; others are computationally-supported frameworks or design approaches [47].

Designers often begin with digital 3D models of the target object. To *cast* a 3D model, they must generate a complement (the object negative), and convert it to physical patterns for mold assembly. Examples where software speeds this process are *Stackmold*, which slices the object negative into laser-cuttable slices [49]; while *Metamolds* helps users optimize silicone molds [4].

Computer graphics yields other approaches to 3D-to-2D mapping. *UV Mapping* is the flat representation of a surface of a 3D model. Creating a UV map is called *UV Unwrapping*. While $[X, Y, Z]$ specify an object's position in 3D space, $[U, V]$ specify a location on the surface of a 3D object. UV Unwrapping uses UV coordinate information to create a 2D pattern corresponding to a 3D surface, thus "unwrapping" it [17]. The Least Squared Conformal Mapping (LCSM) UV Unwrapping algorithm is implemented in the popular open source 3D modeling tool *Blender* [18].

Previous work in computer graphics has investigated the decomposition of 3D geometries into geometries that are suitable for CNC cutting [5]. For example, *Axis-Aligned Height-Field Block Decomposition of 3D Shapes* splits 3D geometries into portions that can be cut with 3-axis CNC milling [35]. *D-Charts* converts complex 3D meshes into 2D, nearly-developable surfaces [24].

These tools signified that FoldMold also needed computational support; however, our papercraft-based technology is utterly different. In the FoldMold pattern-generation tool, the 3D model of the object positive is "unwrapped" and elements are added to reassemble – fold – the 2D patterns into a structurally sound mold. Similar algorithms have been used in other tools, such as the Unwrap function in Fusion 360 [2], but not in a prototyping or mold-making context.

## 3 THE FOLDMOLD TECHNIQUE

In its entirety, FoldMold is a fast, low-cost and eco-friendly way to cast objects based on a 3D digital positive. It can achieve an identifiable set of geometries (Section 3.1), utilizes a set of mold design features (3.2), and consists of a set of steps (3.3). In Section 4 we describe our custom computational tool (*FoldMold Pattern Builder*) which makes designing complex FoldMolds feasible and fast.

### 3.1 FoldMold Geometries

A flexible piece of paper can be bent into many forms. Termed *developable surfaces*, they are derivable from a flat surface by folding or bending, but not stretching [46]. Mathematically, such surfaces possess zero Gaussian curvature at every point; that is, at every point on the surface, the surface is not curved in at least one direction. Cylinders and cones are examples of curved developable surfaces, but spheres are not: every point on a sphere is curved in all directions.

FoldMold can be used for any developable surface or connected sets of them. It can achieve a non-developable surface after approximating and translating it into a subset of connected, individually developable surfaces (*islands*), which can then be joined together. A single developable surface may also be divided into multiple islands, *e.g.*, for ease of pattern construction or use. Islands comprise the basic shapes of a 2D FoldMold pattern (Figure 1, Step 3).

FoldMold geometries can have several kinds of *edges. Joints* are seams between islands. *Folds* (sharp, creased bends) and *smooth curves* are both controlled via scoring, *i.e.*, cuts partway through a material, possible with a lasercutter or handheld knife.

A FoldMold island can have multiple *faces* which are equivalent to their 3D digital versions' polygons, *i.e.*, the polygon or face resolution can be adjusted to increase surface smoothness. Faces are delineated by any type of edge, whether cut or scored.

A strength of the FoldMold technique is its ability to construct *large* geometries. The size of a FoldMold geometry is characterized by three factors.

*Size/time scaling:* While popular 3D printers accommodate objects of 14-28cm (major dimensions), build time scales exponentially with object size. In contrast, FoldMold operations (2D cutting and folding) scale linearly or better with object size (Table 2).

*Weight of cast material:* Paper is flexible, and may deform under the weight of large objects. As we show in 5.2, we tested this technique with a large object cast from plaster (3.64 kg) and did not notice visible deformation. As objects get even larger and heavier, they will eventually require added support.

*Cutter specs:* The cutter bed size limits the size of each island in the geometry. Additionally, the ability of the cutter to accommodate material thickness/stiffness is another limiting factor.

### 3.2 FoldMold Features

FoldMold produces precise, curved, but sturdy molds from paper via computationally managed bending, joinery and mold supports. Here we discuss the features of FoldMold.

**Score-Controlled Bending for 3D Shapes from 2D Patterns**

*Folds (Sharp Creases):* Manual folding can produce uneven or warped bends, especially for thick or dense materials. To guide a sharp fold or crease, we score material on the outside of the fold line to relieve strain and add fold precision.

*Smooth Unidirectional Curves:* As is well-known by foam-core modelmakers, repetition of score lines can precisely control a curve. For example, as we add lengthwise scores on a cylinder's long axis, its cross section approaches a circle; non-uniform spacing can generate an ovoid or U-shape. There is a trade-off between curve continuity, cutting time and structural integrity. Designers can adjust scoring density – the frequency of score lines – based on specific needs; *e.g.*, speed often rules in early prototyping stages, replaced by quality as the project reaches completion. We can smooth some discretized polygonization by filling corners and edges with wax.

**Joinery to Attach Edges and Assemblies**

Joints must (1) seal seams, (2) maintain interior smoothness, for casting surface finish, and (3) support manual assembly. We implemented sawtooth joints, pins, and glue tabs (Figure 2).

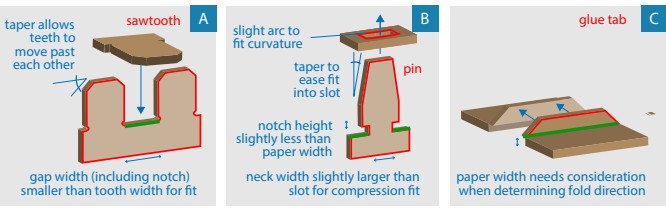

Figure 2: *FoldMold joint types* (A) *Sawtooth* and (B) *pin* joints utilize pressure fitting for secure joints and to maintain alignment. (C) Glue tabs rely on an adhesive.

*Sawtooth Joints:* Pressure fits create a tight seal, with gaps slightly smaller than the teeth and held by friction, enabled by paper's compressibility (as shown in Fig. 2A). To ease insertion, we put gentle guiding tapers on the teeth, with notches to prevent pulling out. Best for straight, perpendicular seams, these joints can face outward from the model for interior surface integrity.

*Pin Joints:* Small tabs are pushed through slightly undersized slots; a flange slightly wider than the corresponding slot ensures a pressurized, locking fit (Fig. 2B). Pin joints are ideal for curved seams that other techniques would discretize: *e.g.*, a circle of slots on a flat base can smoothly constrain a cylinder with pins on its bottom circumference. Tapers and notches on the pins facilitate assembly.

*Glue Tabs:* Fastest to cut and easiest to assemble, two flat surfaces are creased then joined with adhesive (Fig. 2C). Overlapping the tabs (as in cardboard box construction) would create an interior discontinuity. Instead, we bend both tabs outwards from the model and paste them together, like the seam of an inside-out garment. Thus accessible, they can be manipulated to reduce mismatch while preserving interior surface quality.

*Ribbing for Support:* Wax stiffening greatly strengthens the paper, In some cases, *e.g.*, for dense casting materials such as plaster, or large volumes, more strength may be needed to prevent deformation. External support can also help to maintain mold element registration (Figure 1, Steps 2 and 4).

### 3.3 Constructing a FoldMold Model

Constructing a mold using the FoldMold method can be described in five steps, illustrated in Figure 1. Section 4 describes how the *FoldMold Pattern Builder* ("Builder" hereafter) assists the process.

**Step 1: Create or import a 3D model.** The designer starts by modelling or importing/editing the 3D object positive in Blender.

**Step 2: Design the FoldMold – iteratively add and refine seams, bends, joinery and supports:**

*Substeps:* Conceptually, FoldMold design-stage subtasks and outputs are to (a) indicate desired joint lines (island boundaries) on the 3D model (b) joinery type for joints; (c) adjust face resolution to achieve desired curvature; (d) specify scoring at internal (non joint) face edges to control bending within islands, and (e) add ribbing for

mold support. Each of these tasks are supported in the FoldMold Pattern Builder tool's interface (below). Any of these substeps may be repeated during digital design, or revisited after the mold has been physically assembled to adjust the design. This may be especially important for novice users or for challenging projects.

*Automation and Intervention:* Builder can do Step 2 fully automatically, but because its unwrapping algorithm does not consider all factors of the molding process such as preferred building process or seam identification, results may sometimes be improved with maker intervention. As examples, one can intervene at (a) by constraining joint lines then letting Builder figure out scoring (c). By default, Builder uses glue tabs for joints, but we can step in at (b) in a realization that a pin joint will work better than glue tabs for a circular seam such as a cup bottom. Builder's default ribbing is 3 ribs along each of the X- and Y-axes, and 2 Z-axis ribs holding them in place. The maker can intervene to modify ribbing placement frequency, and to position and orient individual ribbing pieces to best support a given geometry.

**Step 3: Unfold and cut the FoldMold design.** Builder unfolds the object's geometry into a 2D mold pattern that is cutter-ready, exported as a PDF file. The maker cuts the 2D patterns from paper by sending the PDF to the cutter – *e.g.*, a laser cutter, vinyl cutter, or even laser-printing the patterns and cutting them with an X-Acto knife or a pair of scissors.

**Step 4: Assemble, wax and cast.** The FoldMold physical construction steps are shown in Figure 3. The maker assembles the cut patterns into a 3D mold by creasing and bending on fold lines and joining at seams according to the joinery method.

To build mold strength, the maker repeatedly dips it in melted wax (paraffin has a melting point of 46–68 °C). As the wax hardens, it stiffens the paper, "locking in" the mold's shape. For very fine areas, dipping may obscure desired detail or dull sharp angles; wax can be added with a small brush, and excess can be removed or surface detail emphasized.

Curable casting materials (*e.g.*, silicone or epoxy resin), or materials that dry (plaster, concrete) are simply prepared and poured.

**Step 5: Set and remove Mold.** After setting for the time dictated by the casting material, the mold is easily taken apart by gently tearing the paper and peeling it away from the cast object. Any excess wax crumbs that stick to the object can be mechanically removed or melted away with a warm tool.

## 4 COMPUTATIONAL TOOL: FOLDMOLD PATTERN BUILDER

A designer should be able to focus effort on the target object rather than on its mold, and FoldMold-making requires complex and laborious spatial thinking, especially for complex shapes. Fortunately, these operations are mathematically calculable, and features can be placed using heuristics. To speed up the mold-making process, our computational tool – the FoldMold Pattern Builder – automates the generation of laser-cuttable 2D patterns from a 3D positive while allowing designer intervention. We describe its implementation and usability evaluation.

### 4.1 Implementation

We wrote Builder as a custom Blender add-on, using Blender's Python API.

#### 4.1.1 User Interface

We created Builder's user interface to reflect primary FoldMold design activities, as described in Section 3.3. The interface's panels shown in Figure 4 map to Steps 2a-d (panel A, *Mold Prep*), Step 2e (panel B, *Ribbing Creation*) and Step 3 (panel C, *Mold Unfolding*).

Builder's user interface is integrated into the Blender user interface, following the same style conventions as the rest of the software in order to reduce the learning curve for novice users who may

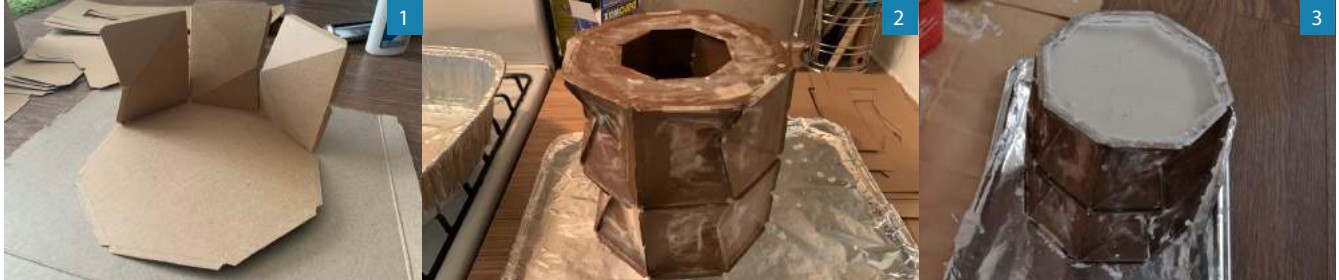

Figure 3: *FoldMold physical construction* (1) Assemble the mold: in this case, tabs are glued and dried. (2) Wax: the mold is dipped in wax strengthen and seal it, preparing it for casting. (3) Cast: the casting material, in this case plaster, is poured into the mold and left to harden.

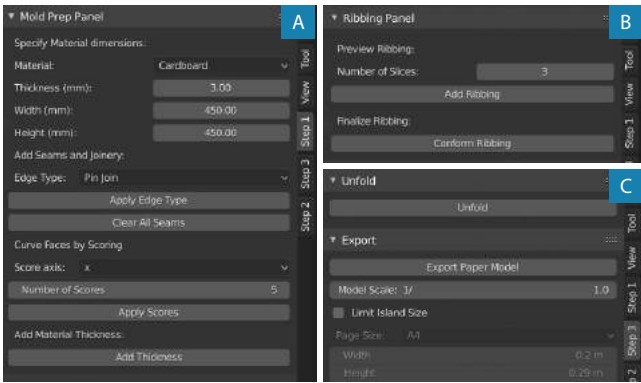

Figure 4: The FoldMold Pattern Builder's user interface has panels based on three design activities, each accessed from a menu bar on the side of the Blender screen. Target model edges and vertices can be first selected while in Blender "edit" mode. (A) *Mold Prep Panel*: Specify material, apply seams and joinery types, and create scores. (B) *Ribbing Panel*: Generate and conform ribbing elements, with user-specification of frequency of ribbing elements. (C) *Unfold Panel*: Export the mold into 2D patterns.

already be familiar with 3D modeling programs. We tested multiple different configurations of the panels before finding that this grouping of options was the most intuitive.

#### 4.1.2 Bending

In contrast to cut and joined seams, bending edges (for both sharp creases and smooth curves; Section 3.1) remain connected after unwrapping and need no joinery; however, they need to be scored.

Builder automatically detects folds as non-cutting edges that demarcate faces, and on its own, would direct a scoring cutting pattern for them (a single score on each of these edges). No scores are placed on the faces by default. As noted in Section 3.3, the user can intervene in a number of ways. Scoring can be applied by following the steps described in Figure 5, of (1) face selection, (2) axis choice from Cartesian options, (3) assigning score density (polygon resolution), and finally (4) applying scores to the faces with the press of a button.

A finely resolved curve can be achieved by adjusting the score density along faces in the 3D object. If a score density has been set (Figure 5, Step 3), Builder creates additional fold lines across those faces beyond its default.

To instruct the cutter how to handle them, Builder assigns colors to cut and fold lines (red and green respectively; Figure 1, Steps 2-3). In the exported PDF, this is a coded indicator to the CNC cutter to apply different power settings when cutting, recognized in the machine's color settings.

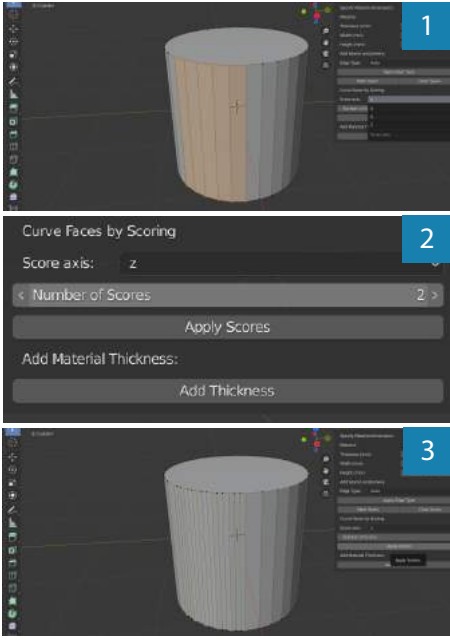

Figure 5: *Creating scores with the FoldMold Pattern Builder.* (1) Select faces of the object that are to be scored. (2) Choose axis around which scores should be drawn and define scoring density. (3) Apply scores to faces.

#### 4.1.3 Joinery

To reassemble the islands created during UV-unwrapping into a 3D mold, Builder defines joinery *features* (sawtooths, pins, glue tabs) along the 2D cut-outs' mating edges in repeating, aligned *joinery sequences*. All three joint types can be included in a given model.

Builder can choose cutting edges. Its default joint type is a glue tab, the easiest to cut and assemble. The user can override this in the Builder interface (Figure 6), instead selecting an edge, then choosing and applying a joinery type.

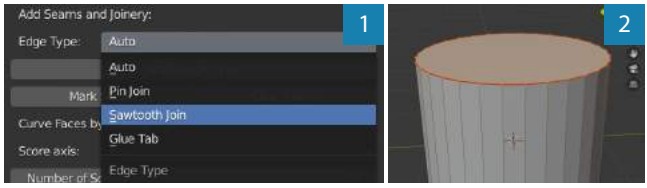

Figure 6: *Creating joinery with the FoldMold Pattern Builder.* (1) Select edges of the object to be joined and choose joinery type or default to "Auto", which are glue tabs. (2) Apply joinery to selected edge.

Builder implements this functionality as follows. The basic com-

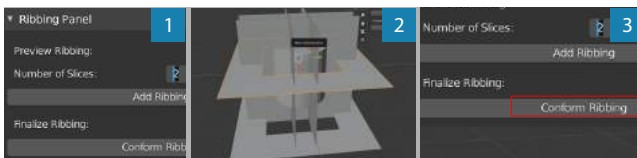

Figure 7: *Creating ribbing with the FoldMold Pattern Builder.* (1) Set number of slices to add along each axis. (2) Add ribbing to object and optionally transform slices around object for maximal support. (3) Select "Conform Ribbing" to finalize the ribbing shape.

ponents of each joinery sequence are referred to as *tiles*, which are designs stored as points in SVG files. Builder defines joinery sequences from several tiles, first parsing their files. This system is easily extended to more joint types simply by adding new SVG images and combining them with existing ones in new ways.

As an example, a sawtooth joinery sequence is composed of tooth and gap tiles. These are arranged in alternation along one edge, and in an inverted placement along the mating edge such that the two sets of features fit together (*i.e.*, *register*). Builder generates a unique sequence for each mating edge pair because the number of tiles placed must correspond to the length of the edge, and matching edges must register, *e.g.*, with pin/holes aligned.

Once created, a joinery sequence must be rotated and positioned along its mating edges. This is automatically done by Builder through transformations (rotation and translation) to each tile sequence to align it with its target edge, and positioning the sequence between the edge's start and end vertices.

### 4.1.4 Ribbing

Builder defines ribbing along three axes (Figure 7) for maximal support and stability. X- and Y-axis ribs slot together, supporting the mold, while Z ribs slot around and register the XY ribbing sheets.

It is impossible to physically assemble ribbing that fully encloses a mold, as the mold would have to pass through it. Builder splits each ribbing sheet in half, then "conforms" it by clipping the ribbing sheets at the mold surface – performing a boolean differenc operation between each ribbing sheet and the mold. In assembly, the user will join the halves to surround the mold.

Within the Builder interface, the user can modify the default ribbing by choosing and applying a ribbing density in terms of slices to be generated by axis (Figure 7). The ribs can then be manipulated (moved, rotated, scaled) within the Blender interface to maximise their support of the object, then conformed with a button click.

### 4.1.5 UV Unwrapping

Conversion of the 3D model into a flat 2D layout is entirely automated through UV Unwrapping (Section 2.4).

At this stage, the Blender mesh object (the 3D model) is converted into a 2D "unfolded" pattern. Our implementation draws from the "Export Paper Model from Blender" add-on [16] from which we use the UV unwrapping algorithm which employs Least Squares Conformal Mapping (LSCM). Initially, the 3D model is processed as a set of edges, faces, and vertices. We then reorient the faces of the 3D object, as if unfolded onto a 2D plane; then alter the edges, faces, and vertices to be in a UV coordinate space.

Unwrapping delivers a set of islands (Section 3.1) – themselves fold-connected faces delineated by joint edges. While these seams are automatically generated during unwrapping, they can optionally be user-defined through Builder's Unfold panel (Figure 4). LSCM minimizes deformation when unfolding and preserves local angles. While it does not guarantee developability in highly complex geometries, a user can adjust seam definitions to create a developable mapping.

If each island exceeds page size (set by media or cutter workspace), it will be rotated to better fit; failing that, the user

Table 1: Mold design time: participant expectations vs. actual time for time for mold design.

| Participant | Estimated 3D-printable mold | Estimated FoldMold (no Builder) | Estimated Fold-Mold (Builder) | Actual FoldMold (Builder) |
|---|---|---|---|---|
| **P1** | several hours | most of a day | **2 min** | **5 min 40 s** |
| **P2** | 30-60 min | several hours | **a few minutes** | **5 min 10 s** |
| **P3** | 30-40 min | 3 hours | **2 min** | **9 min 30 s** |

can then scale the 3D model or define more seams, for more but smaller islands.

### 4.2 Usability Review of FoldMold Pattern Builder Tool

We conducted a small (n=3) user study for preliminary insight into the designers' expectations and experiences with Builder. This study was approved by our university ethics board (certificate number H13-01620-A021).

#### 4.2.1 Method

We recruited three participants (all male Computer Science graduate students whose research related to 3D modeling for familiarity with relevant software). P2 was moderately experienced with Blender, P1 had used Blender, but was not experienced with it, and P3 had never used Blender, but had extensive experience with similar software (3D Studio Max). Conducted over Zoom, sessions took 45–60 min, with the participants accessing Blender and the Builder add-on via Zoom remote control while the researcher recorded the session. Participants were compensated $15.

We introduced each participant to the FoldMold technique, demonstrating how to design joints, bends, and ribbing for various geometries. The researcher walked the participant through a Builder tutorial with a simple practice object (a cube) to design a mold for, and answered questions. They then estimated how long they would take to digitally design a mold for the object in Figure 9, before actually designing and exporting the mold pattern for the object using Builder. The session finished with a short interview.

#### 4.2.2 Results

We review participants' qualitative and quantitative responses to our three questions.

**1. Time: How much time do users expect to and actually spend on mold design?**
All participants predicted Builder would be much faster (2 or a few min) than either designing a 3D-printable mold or manually creating a FoldMold design (30m – a day) (Table 1). Their actual recorded Builder-facilitated times were under 10 minutes (average 6:46). P3, with previous casting experience, iterated on their original design with considerations of manual assembly; their longer time resulted in a slightly more easily assembled mold. While P1 and P2 had no previous mold-making experience, Builder successfully guided them through the creation of a simple mold.

**2. Outcome: Could they customize; and could outcome control be improved?**
All participants reported good outcome control, but offered three possibilities for improvement.

*Joinery density:* Participants tended to select all edges in a curve, applying a joint type to the entire selection. P2 was interested in selecting a set of edges and applying joints to, *e.g.*, every third edge, to prevent overly dense joints on a scored object and consequent assembly complication.

*Mold material combinations:* While Builder allows users to select or define a mold material type (*e.g.*, chipboard or cardboard), they would have liked to indicate multiple materials for a single mold. P3 attempted to define different material settings for different object

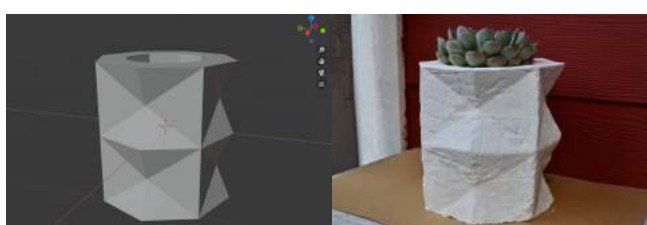

Figure 8: Left: the 3D model of the planter. Right: the physical planter cast with plaster and a plant inserted.

faces, to accommodate regions which needed flexibility (thin bendy paper) versus strength (thick and dense).

*Cut positioning:* When Builder currently exports mold patterns, it arranges pieces to maximize paper usage. P1 would have valued grouping pieces based on relationship or assembly order.

### 3. Problems: What obstacles were encountered?

While participants were generally positive, there were instances where transparency could have been better. P3 was unsure of how Builder would automate mold design without user input, and had to do a trial export to learn it. P1 and P2 asked for warnings when their design choices would lead to issues with the mold or cast object.

## 5 DEMONSTRATIONS

To demonstrate FoldMold performance in our goals of curvature, large scale, and deterministic outcome, we used Builder to design molds for three objects and constructed them in a home workshop. We used a Silhouette Cameo 4 vinyl cutter [1] to cut patterns onto chipboard paper, which we then assembled, dipped in paraffin wax, and cast. We purchased chipboard from an art store ($2.20 / 35x45in sheet), and paraffin from a grocery store at ~$10 per box.

In Table 2 we compare FoldMold construction times for each demo object (from digital design to de-molding, not including material curing) to the time it would take to 3D print the object positive, and the time it would take to 3D print a mold (negative) for casting the object. Mold design and construction for all objects were done by the authors, with their relative expertise with this new technique. The times for each mold construction are taken from a single build. We can see that FoldMold accomplishes much faster speeds, especially as the model size increases.

### 5.1 Curvature

To demonstrate FoldMold curvature and complexity performance, we chose a heat-protective silicone kitchen grip with multiple curvature axes and overhangs which make it harder to 3D print and should also challenge a fold-based technique.

Figure 1 shows the steps for building a heat protective silicone kitchen gripper, beginning by (1) modelling the geometry in Blender.

We (2) scored curved areas, marked mold seams and set joinery types using Builder. The model's varying surface topology indicated a mix of joinery types. For curved areas we chose pin joints, and for all non-curved areas we used glue tabs to keep the interior of the seams flat and smooth. In ribbing design, we chose a ribbing density of one slice per axis in order to keeping the inner and outer edges of the cavities registered, without requiring very much support.

After (3) exporting the mold layout and cut it from paper using our vinyl cutter, we (4) assembled the mold, dipped it in wax, and poured the silicone. Once the silicone had cured, we (5) removed the cast object from the mold.

Mold materials for the gripper mold (excluding casting material and vinyl cutter) cost ~$4.50.

### 5.2 Scale

A FoldMold strength is creating large molds (Section 3.1) without the same speed-size tradeoff common with other rapid prototyping techniques. We demonstrate this by casting a planter that measures 18.8 cm in total height and 17.8 cm in diameter, with an intricate angular outer surface, with a hollow interior to allow for a plant to be inserted (Figure 8) We used plaster for strength.

Mold creation, shown in Figure 3, was similar to Figure 1, with minor adjustments. Due to its angular geometry, the mold did not need to be scored. The long, straight edges could be largely connected using glue tabs and adequately secured with wax. Planter mold materials cost ~$5.50.

### 5.3 Variability

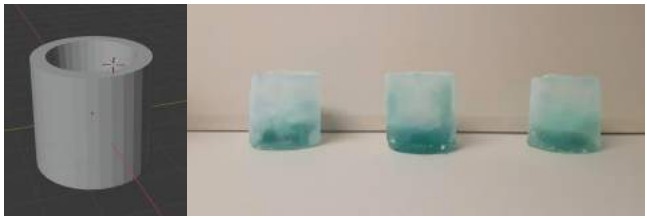

Figure 9: Left: the 3D model of the drinking cup. Right: three physical cups cast in ice.

While rapid-prototyping workflows do not usually involve multiple re-casts of the same object, we wanted to test the extent to which the output is deterministic. In early prototyping stages, it can be beneficial to introduce some variability as a catalyst to ideation and inspiration, whereas in later stages of prototyping, higher determinism is useful as the design approaches completion.

We tested FoldMold's variability by making three cups from the same mold pattern and casting them with ice (Figure 9), following a similar process to that of Figure 1. Due to these molds' small size, ribbing was not needed. The cups' cylindrical geometry led us to use pin joints around the top and bottom, with glue tabs connecting the sides. Table 5 shows the dimensions of each cast cup. Mold materials for each cup cost ~$2.

## 6 DISCUSSION

We review progress towards our goals of accessibility, performance, usability and customisability.

### 6.1 Accessibility: Resource Requirements, Cost, Ecological Load, Versatility

We set out to establish a process that was not just fast, but could be done in a home kitchen (many makers' "pandemic workshop") with readily available, low-cost materials and without toxic waste.

Paper and wax materials together cost $3–$10 per model of the scales demonstrated here, and are easy to source in everyday consumer businesses. Other costs to this project include a computer to design molds, a cutter and casting material. The latter is highly versatile; FoldMold can potentially cast anything that sets at a temperature low enough to not melt the wax, including many food-safe items (we have tried chocolate and gelatin as well as ice).

The disposable mold is biodegradable. We found the mold making materials easy to cut and assemble using a vinyl cutter (a small consumer CNC device). While we could not demonstrate laser-cut examples due to COVID-19 access restrictions, laser-cutters are common in staffed school and community workshops; although more expensive, they avail higher precision and speed.

### 6.2 Speed and Outcome

We targeted fast creation of single-use molds for diverse casting materials in an accessible setting. We compared FoldMold with the go-to of 3D printing (as opposed to other DIY casting methods like StackMold) as it can also achieve geometries we sought.

Table 2: Mold making times, excluding curing (Silhouette Cameo 4 Vinyl Cutter [1]). 3D printing estimates were generated by the Cura Lulzbot 3D printing software [3] at $100mm/s$ printing speed and $1.05g/cm^3$ fill density. We estimate that if laser-cut, FoldMold cuts would be 2-4 times faster.

| Demo | Digital Design | Cutting | Mold Prep & Casting | De-molding | Total FoldMold | 3D Print, Positive | 3D Print, Mold |
|---|---|---|---|---|---|---|---|
| **Kitchen Grip** | 10 min | 41 min | 2 hours | 1 min | **2h 52 min** | 21h 7 min | 41h 13min |
| **Planter** | 10 min | 43 min | 1h 37 min | 1 min | **2h 31 min** | 47h 38 min | 56h 12min |
| **Cup** | 5 min | 22 min | 36 min | 5 min | **1h 8 min** | 5h 20 min | 6h 56 min |

Table 3: Comparing dimensions of digital and cast grips

| Prototype | Height | Width | Depth |
|---|---|---|---|
| *3D model* | **11 cm** | **17 cm** | **10 cm** |
| **Silicone casting** | 11.5 cm | 17.1 cm | 10.5 cm |

Table 4: Comparing dimensions of digital and cast planters

| Prototype | Height | Diameter | Depth |
|---|---|---|---|
| *3D model* | **19 cm** | **18 cm** | **15 cm** |
| **Plaster casting** | 18.8 cm | 17.8 cm | 15.2 cm |

Table 5: Comparing dimensions of three ice cups

| Prototype | Height | Diameter | Thickness | Depth | Capacity |
|---|---|---|---|---|---|
| *3D model* | **7.6 cm** | **7.3 cm** | **1 cm** | **4.8 cm** | **110 mL** |
| **1** | 7.3 cm | 6.9 cm | 0.8 cm | 4.5 cm | 91 mL |
| **2** | 7.5 cm | 6.9 cm | 0.7 cm | 4.8 cm | 93 mL |
| **3** | 7.0 cm | 7.1 cm | 1.0 cm | 4.7 cm | 90 mL |

We found FoldMold build-times to be extremely competitive with 3D printing (Table 2), and that the process is capable of a highly interesting range of shape and scale at a fidelity level and surface quality that makes it a viable casting alternative. With 3D printing, a maker faces fewer steps but will wait longer for their mold or positive to print, particularly for larger objects. Our FoldMold planter took 2.5 hours to cut, assemble, and dip in wax. This would have taken a 3D printer 48 hours for a 3D positive and 56 hours for the mold.

Compared to 3D printing an object positive or mold, FoldMold requires a more hands-on approach; and maker skill (mold design optimizations and mold-craft "tricks") can improve results. However, users already find the current process straightforward.

Beyond efficiency, mold-handling and shape "tweaking" are an opportunity for spontaneous, fine-grained control over the final geometry beyond what is captured in the digital model. Techniques that rely on the removal of material, such as folding or scoring, may leave the surface finish with unwanted ridges; wax dipping prevents this, filling and smoothing cuts. Finally, wax-soaked paper is a convenient non-stick surface that is easy to remove.

We aimed to support creation of highly curved surfaces. Computational support removed most limits: FoldMold should be able to achieve anything within the space of 1D curves at some scale. For *non-developable* surfaces (entirely or as a part of a hybrid mold) or highly precise geometries, 3D printing may be more suitable.

Finally, we were pleased by FoldMold's versatility, not only in casting material but in adaptability of the method itself. A mold can be adjusted to tradeoff precision for construction time and material use (more faces and structural elements). Many items can be completely hand-made, albeit more slowly, or the process can be boosted with more powerful tools. We foresee that this technique could be adjusted within itself (*e.g.*, to support multiple paper weights within a FoldMold, as per a study participant's suggestion) but also combined easily with other complementary techniques.

### 6.3 Usability and Customisability

FoldMold's mold design process is facilitated by the FoldMold Pattern Builder. In our user study, participants could design one of our demonstration molds in an average of 6:47 minutes; based on participants' well informed estimations, a mold of the same shape would have taken several hours to design. Cutting down on mold design time is a major benefit of FoldMold.

Alongside Builder's ability to quickly create molds, we aimed to balance user control and tool automation. Our participants were able to digitally customize their molds to assign specific joint types, materials, and structural supports. While customization allows the user to design a mold specific to their making needs, it also offloads intricate design processes by automating the 2D cut patterns of joinery, scoring, and ribbing. Based on participant responses, a desirable adaptation of the tool would account for customizations such as handling different paper material types in one mold or mixed casting materials (*e.g.*, silicone and plaster).

## 7 CONCLUSIONS AND FUTURE WORK

We have contributed a novel paper and wax mold-making technique that allows 3D molds to be constructed from 2D cut patterns. We demonstrated FoldMold's capabilities by demonstrating curvature, scale, precision and repeatability. We developed the FoldMold Pattern Builder, a computational tool that automatically generates 2D mold patterns from 3D objects with optional designer control, dramatically reducing design time from days to minutes. We conducted a small user study to investigate how Builder can better support designers, and found that increasing user control over joinery density, material combinations, and island positioning would be helpful. Here, we discuss the directions that future work should explore.

**Quasi-Developable Surfaces:** FoldMold currently implements only straight-line bends, but like origami, it could employ methods like controlled buckling to achieve curved 3D fold lines for a larger set of geometries. Relatedly, *kerfing* is a woodworking technique in which flat materials are controllably bent in two dimensions via intricate cut-away patterning [10, 22, 31, 51]. Because kerfing *removes* material it can stretch as well as bend, and attain quasi-developable surfaces. Future work should explore how buckling and kerfing can support more complex FoldMold curvatures.

**Assembly Optimizations:** As FoldMolds get more complicated, their hands-on assembly becomes more challenging. Future work should explore ways to computationally optimize the components of the mold for faster assembly. For example, joinery can be minimized and placement of seams optimized. This utility will often be useful in early "draft quality" prototyping stages where model geometries themselves can be simplified for a speed-fidelity trade-off.

**Multi-Material Molds and Casts and Interesting Inclusions:** Multiple molding materials (*i.e.*, different paper weights) can theoretically be used together for molds that are very flexible in certain areas and very strong in others. Certain prototypes may require multiple casting materials in the same mold, and this would influence how the 2D mold pieces fit together and the needed support structures. This can potentially be expanded to support prototyping objects with embedded electronic components – *e.g.*, sensors and actuators for soft robotics and wearable electronics.

## 8 ACKNOWLEDGMENTS

We thank the members of the SPIN Lab and MUX at UBC for their helpful ideas and feedback. We especially thank Qianqian Feng, Tim Straubinger, Bryan Lee, and Liam Butcher for their assistance with the computational tool and material tests. This work was supported by the Natural Sciences and Engineering Research Council of Canada (NSERC) and UBC's Designing for People (DFP).

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
