# OpenReview forum: "FoldMold: Automating Papercraft for Fast DIY Casting of Scalable Curved Shapes"
_graphicsinterface.org/Graphics_Interface/2021/Conference — GI 2021_

### Official Review · AnonReviewer2 · 2021-01-13
**A nice idea but lacking key information about the level of automation in the mold design software**

**Rating:** 6
**Confidence:** 3

**Review:**

This work proposes a new DIY method of mold creation using paper and wax for casting operations. Apart from the idea, the method is augmented using a Blender app where the user can define cuts or perforations such that the mold can take the desired shape.

The strength of this paper is the nice, simple and probably practical idea. I would like to see more examples of casting different objects to support the practicality aspect.

The weakness of this paper is the exposition. I am not talking about writing. The most important question is that how much automation is in the mold design app? Reading the paper, I could not really tell. Furthermore, the texts in Figures 4, 5, 6, and 7 is not readable when I print the article. They can be improved.

References are for the most part good. The work on making clear molds for photopolymers ("FabSquare: Fabricating photopolymer objects by mold 3D printing and UV curing") and on flexible shells ("FlexMolds: automatic design of flexible shells for molding") could be cited.

I think the paper is clearly written and I only spotted a few possible mistakes in writing:
- I think the reference bracket should come immediately after "et al.".
- Section 3.1: "also be divided onto multiple islands" -> "also be divided into multiple islands"
- Section 7, the quotation mark around "draft quality" should be corrected.

---

### Official Review · AnonReviewer3 · 2021-01-13
**Fast but limited mold prototyping**

**Rating:** 6
**Confidence:** 3

**Review:**

The paper presents a system to generate 2d cut patterns of casting molds that can be easily produced and assembled from paper to be used for casting objects.

The idea could be useful for fast prototyping but is very limited in terms of scope and automatisation. The method seems to be only useful for very simple polygonal shapes which excludes most use cases for molding. Even for these simple cases manual intervention seems to be usually necessary.

Maybe I missed it, but I did not see any guarantees for unfoldability. Moreover, using LSCM for unfolding does not guarantee an isometric mapping. As a consequence the mould might not be assemblable from the 2d cut pattern.

While the described process of waxing the paper and fabricating the molds is interesting, I am not convinced that the presented plugin provides a big advantage over manual modelling given the small scale of objects. Therefore I see the contribution mainly in the process itself.

My rating is still leaning towards acceptance as fast prototyping remains relevant and because the idea itself is innovative and could potentially inspire further research.

---

### Official Review · AnonReviewer1 · 2021-01-13
**Developable mold creation with paper and wax**

**Rating:** 6
**Confidence:** 4

**Review:**

The authors present a Blender add-on which allows a user to create a paper-and-wax mold for rapid prototyping. They show their method is fast and relatively cheap for simpler shapes. The resulting models are rough approximations, with the wax covering resulting in some defects, but this may be suitable just for prototyping. A main concern of mine is that the speed advantages would likely decrease with more complex shapes and would require successively more modelling and manual construction skill.

Perhaps the first limitation is that the user must break the model up into developable pieces that approximate the shape. (Actually, the authors suggest that there is an automatic procedure, but do not discuss this method). For shapes with parts that are far from developable, this is a non-trivial modelling task, and may require many individual pieces ("islands," for them). Here it could be useful to perhaps leverage techniques from "Making Papercraft..." by Mitani & Suzuki, or the more recent "Shape Approximation by Developable ..." by Ion et al. to come up with automated techniques.

On a related note, the use of LSCM is a bit strange, as it is attempting to produce a conformal parametrization (or UV unwrapping, as they call it), and would not necessarily preserve developability. I think again, it is left to the user to ensure that this does not become an issue.

On the positive side, I did find the exposition to be fairly clear, and it seems to be a novel approach as far as I can tell. And they've certainly put some effort into making a nice user interface. As such, I'm willing to give it a rating that just passes.

Last note: the pictures for the tool (Figs. 4-7) are all rather low-quality and nearly impossible to read. These should certainly be improved, if the paper is accepted.

---

### Meta-Review · Area_Chair1 · 2021-01-15

**Recommendation:** Accept
**Confidence:** 4

**Metareview:**

The three reviewers seem to be in agreement on nearly all fronts, resulting in a weak accept.

Pros:
* Idea is novel and simple
* Cheap, environmentally friendly method for fast prototyping
* Reasonable exposition, and a nice UI provided

Cons:
* Limited scope for producing complex shapes
* Low automation, likely requiring lots of manual work (splitting into developable pieces, joint specification, scoring for curvature, etc.)
* Developability not guaranteed by their use of LSCM

---

### Decision · Program_Chairs · 2021-01-16

Accept